# Classical conditioning drives learned reward prediction signals in climbing fibers across the lateral cerebellum

William Heffley, Court Hull*

Department of Neurobiology, Duke University School of Medicine, Durham, United States

**Abstract** Classical models of cerebellar learning posit that climbing fibers operate according to a supervised learning rule to instruct changes in motor output by signaling the occurrence of movement errors. However, cerebellar output is also associated with non-motor behaviors, and recently with modulating reward association pathways in the VTA. To test how the cerebellum processes reward related signals in the same type of classical conditioning behavior typically studied to evaluate reward processing in the VTA and striatum, we have used calcium imaging to visualize instructional signals carried by climbing fibers across the lateral cerebellum in mice before and after learning. We find distinct climbing fiber responses in three lateral cerebellar regions that can each signal reward prediction. These instructional signals are well suited to guide cerebellar learning based on reward expectation and enable a cerebellar contribution to reward driven behaviors, suggesting a broad role for the lateral cerebellum in reward-based learning.
DOI: https://doi.org/10.7554/eLife.46764.001

## Introduction

The cerebellum plays a key role in motor control and motor learning. However, it is becoming widely appreciated that the cerebellum also contributes significantly to many non-motor processes (*Schmahmann, 1991*) such as cognition (*Kim et al., 1994*), social processing (*Van Overwalle et al., 2014*), aggression (*Reis et al., 1973*) and emotion (*Schmahmann and Caplan, 2006*). Moreover, cerebellar deficits are associated with non-motor conditions including autism spectrum disorders (*Wang et al., 2014*), deficits of language processing and vocal learning (*Ackermann, 2008*), schizophrenia (*Mothersill et al., 2016*) and other impairments (*Ivry and Spencer, 2004*). Together, these observations strongly suggest that the cerebellum, and cerebellar learning, must operate in a manner compatible with adjusting motor as well as non-motor processes.

Classical models posit that cerebellar learning is instructed by signals carried by an afferent fiber projection from the inferior olive called climbing fibers (*Albus, 1971*; *Ito, 1972*; *Marr, 1969*). Such models suggest that climbing fibers operate according to a supervised learning rule to instruct changes in motor output by signaling the occurrence of movement errors. However, it is unclear how such a supervised learning rule could contribute to modification of many non-motor behaviors. In line with this view, accumulating evidence has demonstrated a wide range of climbing fiber responses that are distinct from motor error signals (*Kitazawa et al., 1998*; *Streng et al., 2017*), including temporal-difference error signals (*Ohmae and Medina, 2015*) and reward prediction signals (*Heffley et al., 2018*; *Kostadinov et al., 2019*; *Larry et al., 2019*). Together, such data provide evidence that the cerebellum may utilize alternative learning rules to support a wider range of behaviors than can be driven by error-based supervised learning alone.

Under what conditions might the climbing fiber system instruct alternative learning rules? Rabies tracing experiments have identified a disynaptic pathway from the lateral cerebellum to the striatum

*For correspondence:
hull@neuro.duke.edu

**Competing interests:** The authors declare that no competing interests exist.

(*Hoshi et al., 2005*), and recent functional connectivity experiments have revealed a direct, mono-synaptic connection from the lateral cerebellum to the ventral tegmental area (VTA) (*Carta et al., 2019*). These experiments also showed that lateral cerebellar output can modulate reward-driven behaviors by the same pathway. If the cerebellum acts to modulate striatal and VTA pathways involved in reward-driven behaviors, it must contain information related to such behaviors, and likely mediate cerebellar learning in a manner consistent with reward-based associations. Indeed, there is evidence that cerebellar granule cells can display reward related signals (*Wagner et al., 2017*), and climbing fibers can display signals consistent with reward prediction errors in certain behaviors (*Heffley et al., 2018*; *Kostadinov et al., 2019*). However, it remains unclear how cerebellar learning, and in particular, how learning signals provided by cerebellar climbing fibers, operate during the types of classical conditioning behaviors classically studied to evaluate reward processing and learning in the VTA and striatum.

A key first step in understanding how cerebellar learning can contribute to reward processing and reward driven behaviors is to test what instructional signals are carried by cerebellar climbing fibers in behavioral paradigms known to involve striatal computations. To this end, we have utilized an appetitive classical conditioning task that mirrors well-studied reward prediction paradigms (*Schultz et al., 1997*), and performed *in vivo* calcium imaging to measure cerebellar climbing fiber driven complex spike (Cspk) activity across the lateral cerebellum in awake mice. These experiments reveal distinct, functionally compartmentalized, reward-related Cspk signals in each of the lateral cerebellar lobules. Specifically, in Lobule Simplex (LS), Crus I and Crus II, robust Cspk responses emerge with learning in response to a visual cue that predicts reward. In LS and Crus II, unexpected reward produces Cspk responses, both before and after learning. In contrast, Crus I shows no Cspk response to unexpected reward in naïve animals, but a robust Cspk response to unexpected reward once animals have associated a visual cue with reward. Imaging Cspk activity across learning suggests that the same climbing fibers that respond to reward in naïve animals can instead respond to a reward predictive cue after learning. Finally, we observe no evidence of negative prediction error following reward omission immediately after learning, suggesting the possibility that Cspk responses may not instruct a strict temporal difference (TD) learning model in this paradigm. The learned, reward predictive Cspk signals observed here are, however, consistent with a model that would allow learned cerebellar output to provide information about rewarding stimuli to downstream structures such as the VTA and striatum. Hence, these data provide a key link between the instructional signals known to be involved in cerebellar learning and the types of reward-based processing observed in downstream areas.

## Results

### An appetitive classical conditioning paradigm to investigate climbing fiber responses in the lateral cerebellum

To investigate climbing fiber driven Cspk activity during an appetitive classical conditioning paradigm that parallels tasks typically used to study reward processing in the VTA and striatum, we first established an appropriate behavior for head-fixed mice that is compatible with calcium imaging in the lateral cerebellum (*Figure 1A–D*). In this task, mice learn a simple association between a visual cue and the timing of subsequent reward delivery. Specifically, mice were trained by viewing a continuously present, full screen vertical grating that was transiently (100 ms) replaced with a horizontal grating to signal the delivery of a saccharine reward at a 600 ms delay (*Figure 1D*). After multiple training sessions (mean = 5.7 ± 0.4 days), mice learned the association between the visual cue and the time of reward delivery, significantly reducing both their reaction times (Day 1, 742.8 ± 25.4 ms; Day N+1, 544.7 ± 13.0 ms; p=8.52 × $10^{-8}$, n = 27 mice, paired t-test, *Figure 1E*) and miss rates (Day 1, 0.39 ± 0.04; Day N+1, 0.03 ± 0.01; p=6.37 × $10^{-9}$, paired t-test, *Figure 1F*). In addition to developing well-timed licking responses that approximated the time of reward delivery on rewarded trials, mice also developed robust licking responses following the visual cue on omission trials after learning that were absent in naïve animals (*Figure 1G*). Together, these data indicate that mice successfully learn the association between visual cue and the timing of reward delivery in this paradigm.

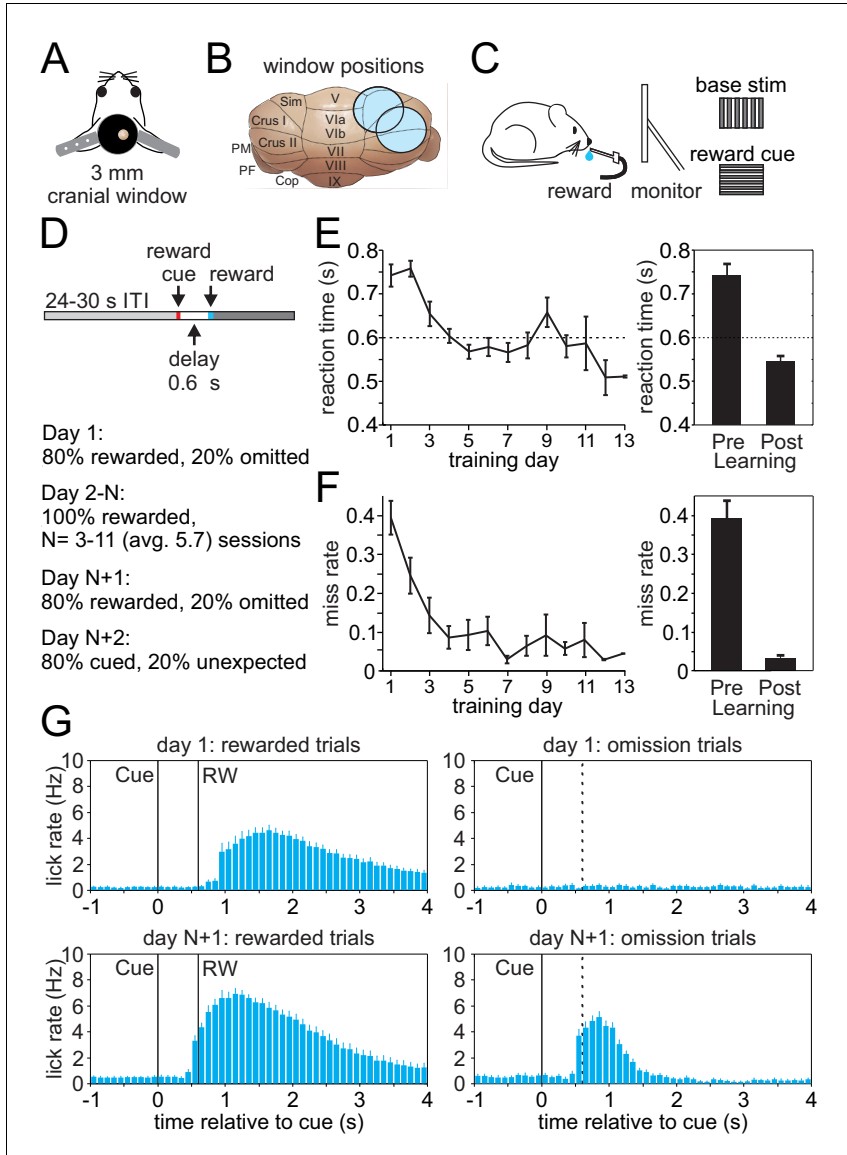

**Figure 1.** Appetitive classical conditioning regime for head-fixed mice. (A) Diagram of headplate and 3 mm cranial window. (B) Scale representation of the 3 mm cranial windows over either Lobule Simplex (LS) or Crus I and II. (C) Mice viewed a static full screen vertical grating that was transiently replaced by a horizontal grating for 100 ms to cue reward delivery following a 600 ms delay. (D) Trial structure and progression of learning sessions. (E) Left, mean reaction time, defined as the first lick following the visual cue, on rewarded trials across training days. Right, mean reaction time on rewarded trials before learning and after learning. (F) Left, mean miss rate, defined as the fraction of trials with no licks within 1 s after the cue, on rewarded trials across training days. Right, mean miss rate on rewarded trials before and after learning (G) Mean lick rates aligned to visual cue onset, before (top) and after (bottom) learning for rewarded trials (RW, left) and omission trials (right, dotted line indicates time when reward would have been delivered).

DOI: https://doi.org/10.7554/eLife.46764.002

## Cspk activity shifts earlier in time after learning

We began by measuring Cspk activity before and after learning in lobule simplex (LS), where previous work has identified reward-related activity in granule cells and climbing fibers after learning in different behavioral paradigms (*Heffley et al., 2018*; *Kostadinov et al., 2019*; *Wagner et al., 2017*). To obtain a broad view of Cspk activity across LS and assess whether any reward-related signals are carried by climbing fibers in this paradigm, we first performed single-photon, mesoscale

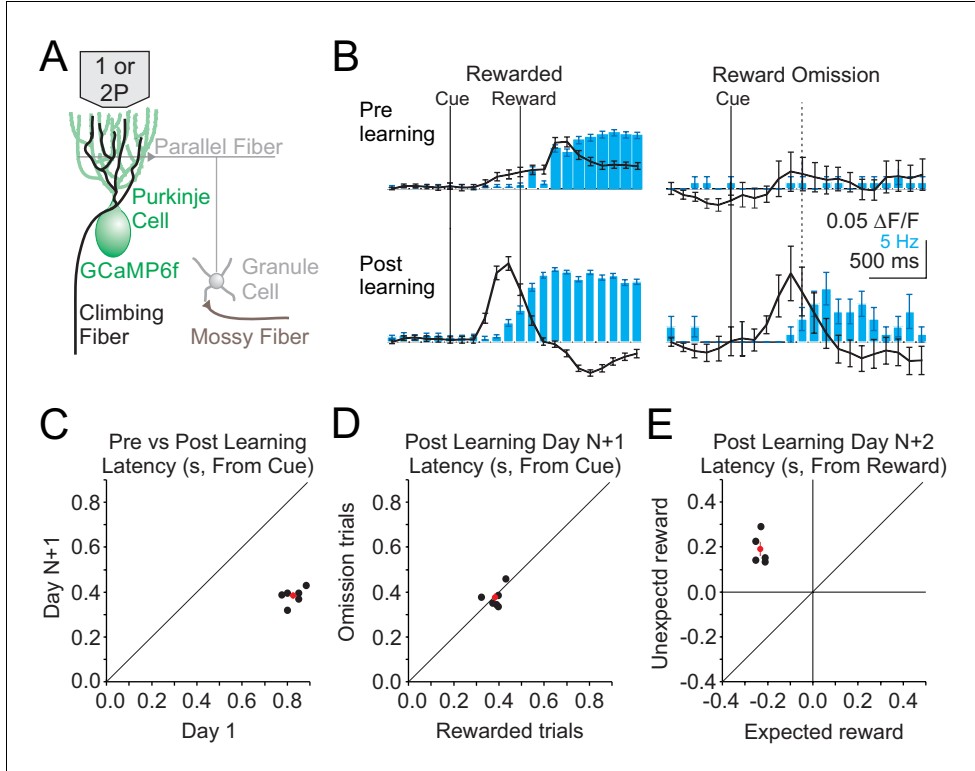

**Figure 2.** Single photon, mesoscale imaging of Cspk activity during behavior. (**A**) Schematic of the cerebellar cortex with GCaMP6f expressed in PCs for single or two photon dendrite imaging. (**B**) Average ΔF/F timecourse (black, collected at 10 Hz) and lick rate (blue bars) for an example animal before (top) and after (bottom) learning for rewarded (left) and reward omission (right) trials. (**C,D,E**) Summary of bulk Cspk response latencies relative to the visual cue (**C,D**) or the reward (**E**) for all trial types. Response latency is defined by the maximum positive change in ΔF/F rate within 500 ms before the peak response.

DOI: https://doi.org/10.7554/eLife.46764.003

calcium imaging of Purkinje cell (PC) dendrites expressing GCaMP6f (*Figure 2A*). This method has previously been demonstrated to accurately report Cspk activity (*Heffley et al., 2018*). Using this approach, we observed a significant increase in the Cspk activity following reward delivery in naïve animals (n = 308 trials, five mice, p=5×10$^{-19}$, paired t-test), but no significant response on reward omission trials (n = 74 trials, five mice, p=0.595, paired t-test) (*Figure 2B*, top). After learning, significant Cspk responses were maintained (n = 749 trials, six mice, p=3.3×10$^{-100}$, paired t-test), but were shifted in time, and preceded reward delivery on rewarded trials (*Figure 2B*, bottom, *Figure 2C*). Moreover, after learning there was also a significant response on omission trials preceding reward delivery (n = 156 trials, six mice, p=5.4×10$^{-16}$, paired t-test) (*Figure 2B*, bottom, *Figure 2D*). Across experiments, the latency of Cspk activity decreased significantly (n = 6 mice, p=1.04 × 10$^{-6}$, paired t-test) (*Figure 2C*). Notably, this shift in Cspk timing ($\Delta_{Cspk}$ = 441 ms) was greater than the change in lick timing ($\Delta_{lick}$ = 185 ms), indicating that the Cspk response did not simply follow the timing of motor output. In addition, we find that Cspk responses after learning had the same timing for rewarded and omission trials (Rewarded trial latency = 383 ± 15 ms, Omission trial latency = 378 ± 19 ms, n = 6 mice, p=0.78 × 10$^{-6}$, paired t test) (*Figure 2D*). In contrast, when reward was delivered unexpectedly after learning (i.e. in absence of the visual cue), Cspk activity followed reward delivery, and was significantly delayed compared with rewarded trials (Day N+2; rewarded trial latency = 366.1 ± 31 ms, unexpected trial latency 793 ± 31 ms, n = 5 mice, p=2.40 × 10$^{-4}$, paired t-test). The latency of Cspk responses on unexpected reward trials was, however, equivalent to the timing of responses to rewarded trials in naïve animals (p=0.38, two-sample t-test) (*Figure 2E*). Together, these data reveal that complex spiking shifts in time as a result of

learning, initially following reward delivery in naïve animals and later preceding reward delivery in trained animals, occurring with the same timing on both rewarded and omission trials after learning.

## Cspk responses in LS signal reward predictions that emerge with learning

Our mesoscale imaging experiments suggest that Cspks in LS exhibit reward-related activity that changes as a function of learning in manner that is consistent with reward expectation. However, to further disambiguate Cspk activity related to reward, reward expectation and licking, and to test whether Cspk activity in individual PC dendrites changes across learning, we next used two-photon imaging to visualize Cspk responses in individual PC dendrites. As in our mesoscale imaging experiments, these data revealed significantly elevated Cspk rates following reward delivery in naïve animals (n = 1040 dendrites, 15 mice, p=$5.5\times10^{-48}$, one-tailed t-test) (*Figure 3A,D*, *Figure 3—figure supplement 1*) but no significant response during this window on omission trials (p=0.98). After learning, we observed significantly elevated Cspk rates prior to reward delivery on both rewarded (n = 906 dendrites, 16 mice, p=$2.3\times10^{-60}$, one-tailed t-test) and omission trials (p=$1.4\times10^{-33}$) that were comparable across trial types (p=0.01, paired t-test; *Figure 3B,C,E*). In addition, post-reward Cspk rates were significantly reduced after learning (p=$2.0\times10^{-19}$, unpaired t-test), with small elevations in spiking (reward trials: p=$3.3\times10^{-4}$; omission trials: p=$6.7\times10^{-6}$; one-tailed t-test) that were equivalent on both rewarded and omission trials (p=0.46; paired t-test). Notably, there were no Cspk responses that were reliably suppressed below baseline after the time of expected reward on omission trials (see Materials and methods). These data indicate that Cspk activity shifts from an association with reward prior to learning to the cue that predicts reward after learning. To test whether unique populations of PC dendrites exhibit Cspk responses to reward and following the visual cue after learning, we identified the subset of PC dendrites with significant post-reward responses (n = 204/906 dendrites; *Figure 3—figure supplement 2*). This small subset of PC dendrites also responded following the visual cue (p=$5.9\times10^{-36}$; one-tailed t-test), indicating that a minority of PC dendrites can exhibit Cspk responses to both reward and the visual cue after learning. These data suggest the possibility that some Cspk responses had not fully transitioned between encoding the reward and the reward-predicting cue immediately after learning. However, the majority of PC dendrites (n = 702/906 dendrites) did not respond to the reward after learning, and instead only responded to the reward-predicting cue (p=$3.9\times10^{-36}$, one-tailed t-test; *Figure 3—figure supplement 2*).

To disambiguate Cspk responses related to reward versus reward expectation, we measured responses to unexpected reward after learning (Day N+2). On these trials, Cspk rates increased significantly after the time of unexpected reward delivery (n = 243 dendrites; six mice, p=$1.6\times10^{-8}$, one-tailed t-test; *Figure 3F*). Moreover, with this additional day of training, there was no longer a significant post-reward response on trials where reward was expected (p=0.8). These data indicate that after learning, Cspk responses in area LS are only reward responsive if the reward is unexpected. In the same experiments, Cspk rates increased significantly prior to reward delivery when reward was expected (p=$1.2\times10^{-12}$, one-tailed t-test; *Figure 3F*). Notably, the PC dendrites that were significantly responsive following the reward-predicting cue (n = 84/243 dendrites), were also significantly responsive after the unexpected reward (p=$2.3\times10^{12}$, one-tailed t-test; *Figure 3G*). This suggests that the same PC dendrites exhibit Cspk responses to the reward-predictive cue and unexpected reward, thus signaling predictions under two different conditions. However, lick timing also differs between these two trial types, and changes across learning. Thus, we also sought to further disambiguate reward related Cspk responses from possible Cspk responses driven by motor output during licking.

To directly assess the role of licking in modulating Cspk rates, we first generated a lick-triggered average of Cspk activity from licks generated during the inter-trial interval (ITI). These data revealed no significant Cspk response to licking (p=0.6; n = 768 dendrites, 13 mice, one-tailed t-test; *Figure 3H*), consistent with previous reports showing a lack of lick-related Cspk activity in area LS (*Bryant et al., 2010*). To further address the contribution of licking to the emergent Cspk responses following the reward-predicting cue after learning, we segregated rewarded trials into subsets according to lick timing. Specifically, we averaged trials with the earliest and latest quartiles of licking. This analysis revealed that the peak of the pre-reward Cspks occurred at nearly the same time relative to the cue on trials with early as compared to late licks (Cspk latency on early trials: 300 ms;

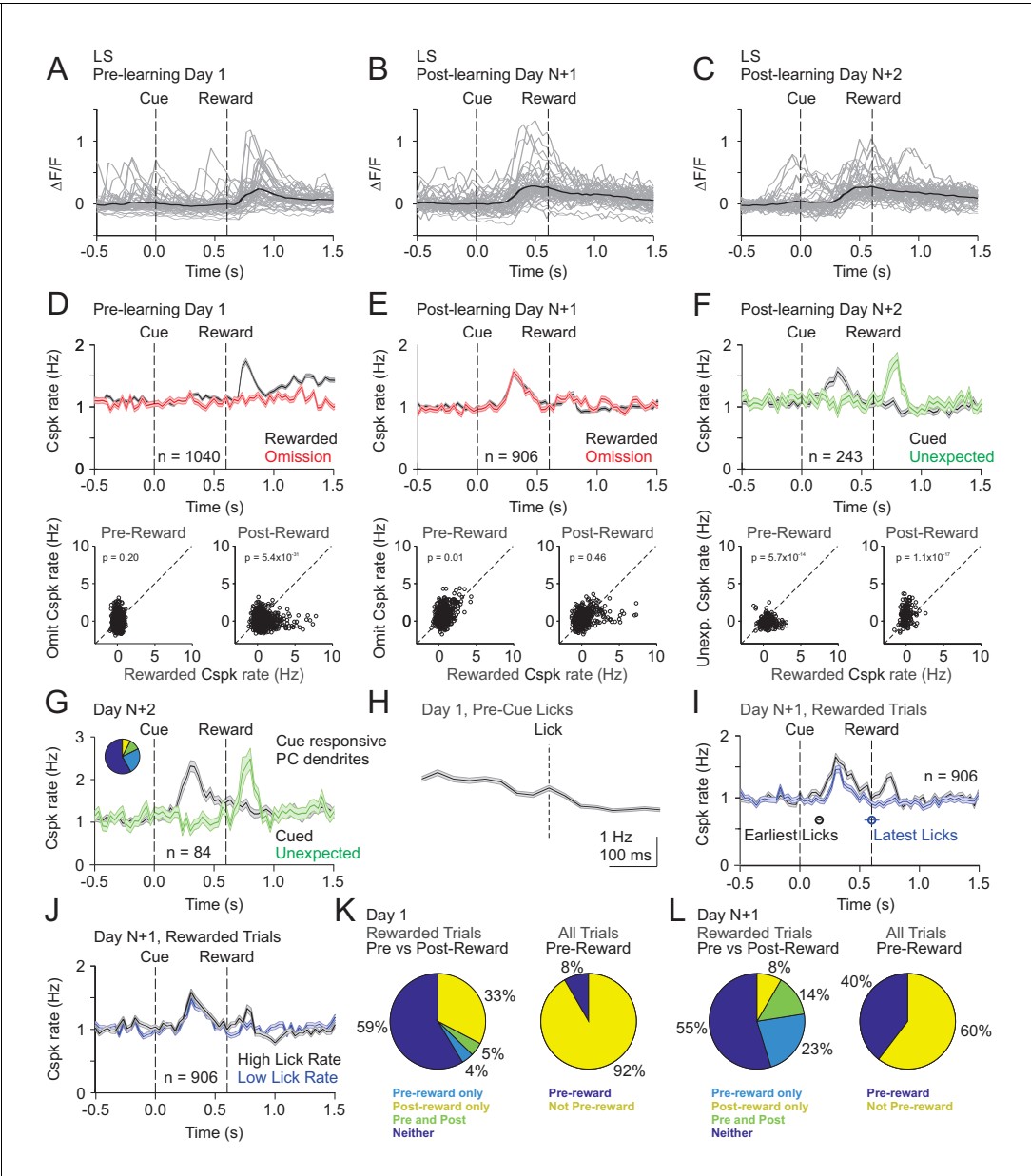

**Figure 3.** Cspk modulation in LS across learning. (A) ΔF/F timecourses from an example neuron measured via two-photon calcium imaging for the first day of training in naïve mice. Gray traces are the first 50 trials and the black trace is the average of those 50 trials. (B) Same as (A), but on post-learning day N+1. (C) Same as (A,B) but for post-learning day N+2. (D) Top, mean cue-aligned peri-stimulus histograms (PSTHs) of Cspk rate on reward (black) and omission (red) trials for all PC dendrites on the first day of training in naïve mice. Bottom, summary scatterplot comparing the Cspk rate for individual PCs on rewarded vs omission trials in naïve mice. Spike rates were measured in a window preceding reward delivery (left) or immediately after reward delivery (right) (Materials and methods). (E) Same as D), but for rewarded and reward omission trials on post-learning day N+1 (F) Same as (D,E), but for rewarded and unexpected reward trials on post-learning day N+2. (G) Mean cue-aligned PSTH for PC dendrites that exhibited Cspk responses to the visual cue. *Inset*: fraction of the total dendrites on Day N+2 that responded in the pre-reward window on rewarded trials only (light blue, 24%), the post-reward window on unexpected reward trials only (yellow, 7%), both pre and post reward windows on rewarded and unexpected reward trials respectively (green, 10%), and neither window (blue, 58%). (H) Mean lick-triggered PSTH for licks during the inter-trial interval (ITI) in naïve animals. (I) Mean cue-aligned PSTHs for trained animals with rewarded trials segregated according to trials with the earlies 1/4 of licks (black, 162 ± 19 ms from cue) and latest 1/4 of licks (blue, 604 ± 64 ms from cue). (J) Mean cue-aligned PSTHs for trained animals with rewarded trials segregated according to trials with the highest 1/4 of lick rates (black, 5.4 ± 0.1 Hz) and the lowest 1/4 of lick rates (blue, 2.6 ± 0.1 Hz). (K) Fraction of all lobule simplex neurons which were responsive to specific task events on the first day of training for rewarded trials (left) and all trials (right). (L) Same as (K) but for training day N+1. Data points with horizontal error bars represent the mean lick timing ± SEM. For all PSTHs, shaded area represents ± SEM across dendrites.

*Figure 3 continued on next page*

*Figure 3 continued*

DOI: https://doi.org/10.7554/eLife.46764.004

The following figure supplements are available for figure 3:

**Figure supplement 1.** Comparison of the mean ΔF/F response for all neurons segregated according to lobule, pre vs post learning, and pre vs post reward delivery. (A-C) (Schematic indicating imaging location (left) and summary of ΔF/F responses before and after learning for each area (right).

DOI: https://doi.org/10.7554/eLife.46764.005

**Figure supplement 2.** A minority of PC dendrites can respond to both reward and the reward-predictive cue after learning.

DOI: https://doi.org/10.7554/eLife.46764.006

**Figure supplement 3.** No evidence of Cspk suppression following reward omission.

DOI: https://doi.org/10.7554/eLife.46764.007

late trials: 330 ms), and with similar, but slightly smaller, amplitude ($p=4.4\times10^{-4}$; n = 906 dendrites, 16 mice, paired t-test; *Figure 3I*). There was, however, a difference in Cspk rate following reward delivery ($p=8.8\times10^{-9}$), likely due to the increased variability in the timing of reward consumption for trials with late licks. In addition, we also compared trials with the highest and lowest lick rates, and found that Cspk responses had identical timing (high and low lick rate Cspk latency = 300 ms; *Figure 3J*) and a qualitatively similar, though statistically different amplitude prior to reward delivery ($p=7.4\times10^{-05}$, paired t-test; *Figure 3J*). These data indicate that the pre-reward Cspk responses that emerge with learning (*Figure 3K,L*) are related to the cue-driven expectation of reward, and not conditioned motor output due to licking.

## The same climbing fibers can respond to reward and a reward-predictive visual cue

While our data indicate Cspk activity within individual dendrites can represent different reward-related events (i.e. the reward-predicting cue and unexpected reward), these data do not address whether it is the same climbing fibers that shift their responses, or whether separate populations of climbing fibers generate different responses before and after learning. To address this question, we measured Cspk activity from PC dendrites in the same coordinates before and after learning in a subset of experiments. To evaluate responses, we took two approaches. First, we manually identified individual PC dendrites across imaging days and measured their Cspk responses before and after learning (*Figure 4A,B*). This approach revealed clear examples of single PC dendrites with reward responsive Cspk activity in naïve animals that became cue responsive after learning. However, because this approach relies on user identification of the same PC dendrites, we also sought to test an automated approach. To do so, we first independently generated pixel masks for PC dendrites from our pre and post-learning fields of view (*Figure 4C–D*). We then motion registered the mean fluorescence image from the post-learning session to the pre-learning session (*Figure 4E*), and applied the same pixel shifts from this registration to the segmented pixel masks from the post-learning condition. By overlaying the shifted post-learning pixel masks with the pre-learning masks, we assessed the overlap between imaging days (*Figure 4F–H*). Despite closely matched fields of view, this process led to a majority of PC dendrite masks with little overlap. We thus employed a stringent criterion of greater than 50% overlap to increase the probability of accurately identifying the same PC dendrites across imaging sessions. With these criteria, we identified a group of PC dendrites likely corresponding to the same PC dendrites before and after learning. In these PC dendrites, as in our other experiments, we find that Cspk activity was initially absent following the visual cue in naïve animals ($p=0.59$; n = 61 dendrites, 14 mice, one-tailed t-test; *Figure 4I*), and that cue driven responses emerged with learning ($p=1.2\times10^{-8}$). We note that because of the close spacing and similar size and orientation of PC dendrites, such approaches cannot provide definitive evidence of tracking the same PC dendrites across learning. However, these data support the conclusion that individual CFs can develop a learned response to the reward predictive cue as a result of learning (*Figure 4J*).

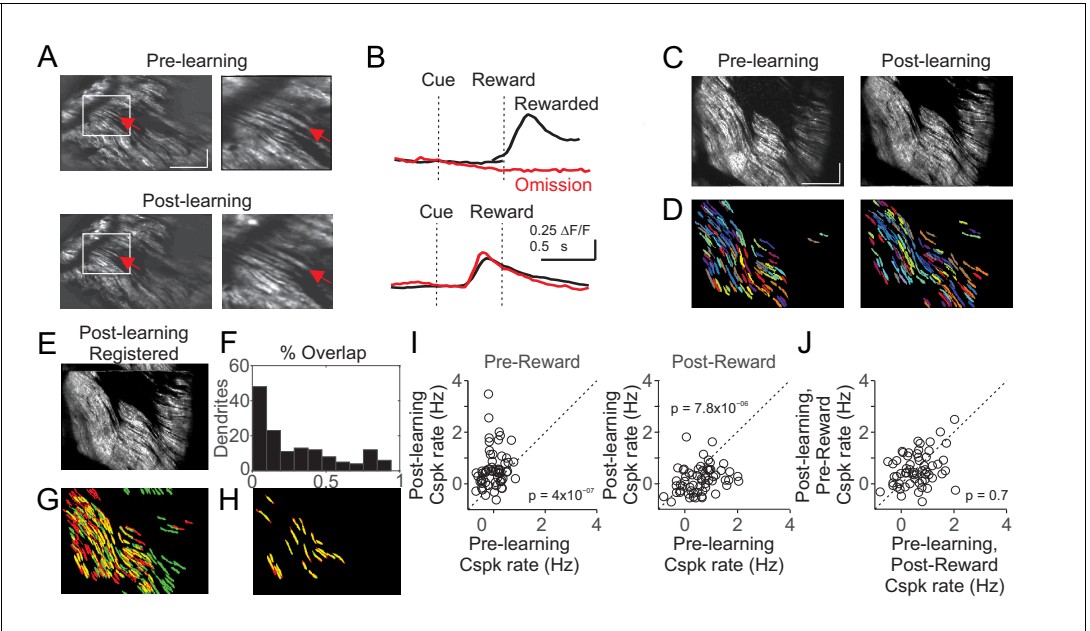

**Figure 4.** Cspk responses measured in the same fields of view across learning. (**A**) Example of manual identification from a field of view containing several of the same PC dendrites on day one and day N+1. Left, fields of view for day 1 (top) and day N+1 (bottom). Right, magnified view from the white box at left. Red arrows indicate an example PC dendrite identified both pre and post learning. Scale bars are 200 µm (x) x 50 µm (y). (**B**) Cue aligned calcium transients averaged across trials extracted from the PC dendrite identified in (**A**) for day 1 (top) and day N+1 (bottom). (**C**) Mean images taken from day 1 (left) and day N+1 (right) after independent rigid motion registration on each dataset. Scale bars are 200 µm (x) x 50 µm (y). (**D**) Pixel masks of PC dendrites independently extracted from the datasets in (**C**) (Materials and methods). (**E**) The post-learning data set from (**C**) right, was motion registered to the pre-learning dataset from (**C**) left. The resulting x-y pixel shifts were applied to the post-learning dendrite pixel masks from (**D**) right. (**F,G**) Following registration, the overlap between pre and post learning pixel masks was compared quantitatively (**F**) and graphically (**G**) with pre-learning masks labeled red, post-learning masks labeled green, and overlap labeled yellow. (**H**) Dendritic masks from (**G**) that had >50% overlap. (**I**) Summary of Cspk firing rates for individual PC dendrites in the pre (left) and post (right) reward window for all dendrites that had a > 50% overlap in their dendritic masks across learning (n = 61). (**J**) Summary of Cspk rates for the same individual PC dendrites in (**I**) measured in the post-reward window on Day one and the pre-reward window on Day N+1.

DOI: https://doi.org/10.7554/eLife.46764.008

## Crus I and Crus II exhibit reward-related Cspk responses that are distinct from area LS

Anatomical and physiological evidence suggests that Cspk responses are functionally compartmentalized across the medio-lateral axis of the cerebellum (*Apps and Garwicz, 2005*), raising the possibility of different computations and diverse reward-related Cspk activity patterns in lobules other than LS. Thus, we next measured Cspk responses in Crus I and Crus II.

In Crus I, pre-learning Cspk activity differed substantially from LS. Specifically, in naïve animals, Cspk activity was sensory-related, and followed the visual cue at a time well before reward delivery or licking on both rewarded (p=3.6×10$^{-20}$, n = 133 dendrites, six mice; one-tailed t-test; *Figure 5A, D*) and omission trials (p=3.4×10$^{-17}$). After learning, however, once the cue had been associated with upcoming reward, Cspk responses to the visual cue were significantly enhanced (p=2.3×10$^{-8}$, unpaired t-test, n = 178 dendrites, seven mice; *Figure 5B,C,E*). In contrast, there were only weak or non-significant reward responses on rewarded or omission trials, either before (rewarded: p=0.002; omission: p=0.21; one-tailed t-test; *Figure 5D,E*) or after learning (rewarded: p=1.0; omission: p=0.84; one-tailed t-test). Again, however, we did not identify any PC dendrites that were reliably suppressed at the time of expected reward after learning. Notably, despite the lack of a Cspk response to unexpected reward in naïve animals, there was a large Cspk response to unexpected reward in the same animals after learning (p=5.1×10$^{-11}$, n = 141, five mice, one-tailed t-test; *Figure 5F*). These data suggest that in Crus I, where Cspk responses are sensory related in naïve animals, activity can also be generated by expectations that are specifically linked to a learned, reward-predictive sensory cue after learning.

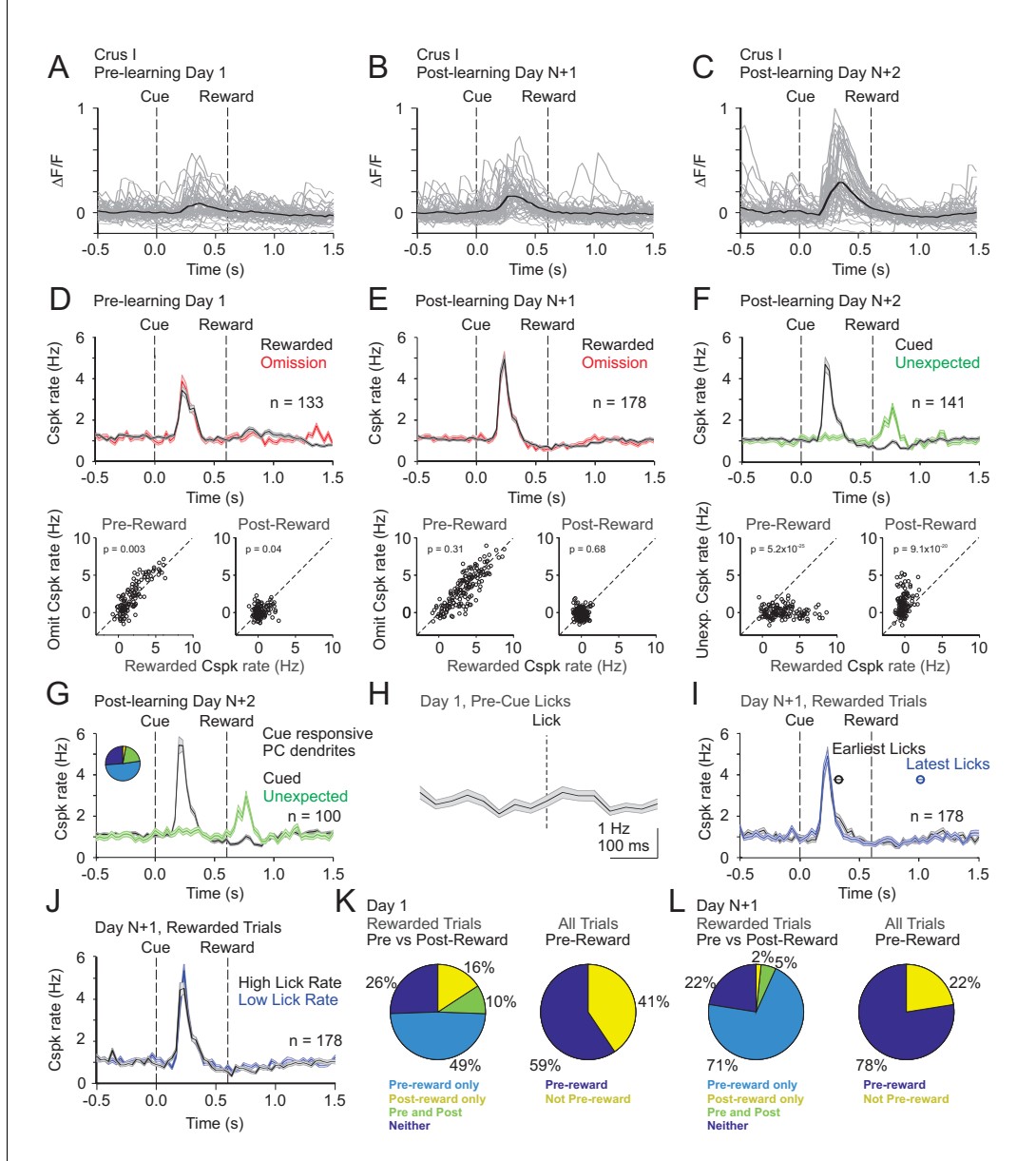

**Figure 5.** Cspk modulation in Crus I across learning. (**A**) ΔF/F timecourses from an example neuron measured via two-photon calcium imaging for the first day of training in naïve mice. Gray traces are the first 50 trials and the black trace is the average of those 50 trials. (**B**) Same as (**A**) but on post-learning day N+1. (**C**) Same as (**A,B**) but for post-learning day N+2. (**D**) Top, mean cue-aligned PSTHs of Cspk rate on reward (black) and omission (red) trials for all PC dendrites on the first day of training in naïve mice. Bottom, summary scatterplot comparing the Cspk rate for individual PCs on rewarded vs omission trials in naïve mice. Spike rates were measured in a window preceding reward delivery (left) or immediately after reward delivery (right) (Materials and methods). (**E**) Same as (**D**) but for rewarded and reward omission trials on post-learning day N +1 (**F**) Same as (**D,E**) but for rewarded and unexpected reward trials on post-learning day N+2. (**G**) Mean cue-aligned PSTH for PC dendrites that exhibited Cspk responses to the visual cue. *Inset*: fraction of the total dendrites on Day N+2 that responded in the pre-reward window on rewarded trials only (light blue, 51%), the post-reward window on unexpected reward trials only (yellow, 3%), both pre and post reward windows on rewarded and unexpected reward trials respectively (green, 20%), and neither window (blue, 26%). (**H**) Mean lick-triggered PSTH for licks during the ITI in naïve animals. (**I**) Mean cue-aligned PSTHs for trained animals with rewarded trials segregated according trials with the earlies 1/4 of licks (black, 326 ± 46 ms from cue) and latest 1/4 of licks (blue, 1010 ± 34 ms from cue). (**J**) Mean cue-aligned PSTHs for trained animals with rewarded trials segregated according to trials with the highest 1/4 of lick rates (black, 4.8 ± 0.1 Hz) and the lowest 1/4 of lick rates (blue, 2.1 ± 0.1 Hz) following reward delivery. (**K**) Fraction of all Crus I neurons which were responsive to specific task events on the first day of training for rewarded trials (left) and all trials (right). (**L**) Same as (**K**) but for training day N+1. Data points with horizontal error bars represent the mean lick timing ± SEM. For all PSTHs, shaded area represents ± SEM across dendrites.
DOI: https://doi.org/10.7554/eLife.46764.009

To test whether the same PC dendrites that exhibit Cspk responses following the visual cue also respond to unexpected reward, we identified those PC dendrites that were significantly cue responsive. Indeed, these PC dendrites (n = 100/141) also showed a robust response to unexpected reward (p=$7.7 \times 10^{-11}$; one-tailed t-test; *Figure 5G*), indicating that PC dendrites can signal expectation under distinct conditions. As further evidence that Cspk responses in Crus I are sensory related in naïve animals, there was no significant Cspk response to licks during the ITI (p=0.04; n = 93 dendrites, four mice, one-tailed t-test; *Figure 5E*). To test whether licking contributes to the learned increase in response amplitude following the cue, we segregated trials according to the timing of the first lick. This revealed that the timing (early and late lick Cspk latency: 230 ms) and amplitude (p=0.04, paired t-test; *Figure 5I*) of Cspk activity following the visual cue was largely independent of lick timing. In addition, we also compared trials with the highest and lowest lick rates, and found the amount of licking also did not alter the timing (high and low lick rate Cspk latency: 230 ms) or amplitude of pre-reward Cspk responses (p=0.4, paired t-test; *Figure 5J*). Thus, as with lobule simplex, learned changes in Cspk responses are not related to conditioned motor output due to licking. Together, these data indicate that Cspk activity in Crus I is sensory related in naïve animals (*Figure 5K*), but can also be modulated by reward expectation because it is enhanced by associating the visual cue with reward (*Figure 5E,L*), and driven by delivering a reward unexpectedly (*Figure 5F*).

Finally, we measured Cspk activity in Crus II. Similar to area LS, Cspk rates were elevated in response to reward delivery in naïve animals (p=$2.7 \times 10^{-18}$; n = 147 dendrites, five mice, one-tailed t-test; *Figure 6A,D*), but not on reward omission trials (p=1.0). After learning, a robust cue response was observed in Crus II on both rewarded (p=$9.7 \times 10^{-26}$; n = 245 dendrites, n = 6 mice, one-tailed t-test; *Figure 6B,C,E*) and omission trials (p=$6.1 \times 10^{-21}$). However, distinct from area LS, the reward response persisted after learning (p=$5.6 \times 10^{-11}$; *Figure 6E*). Again, however, we observed no significantly increased response on omission trials at the time of expected reward (p=0.95); nor did we observe any significant suppression. When reward was delivered unexpectedly after learning, we observed enhanced Cspk rates as compared to the Cspk responses to expected reward (p=$1.1 \times 10^{-11}$, n = 184 dendrites, six mice; *Figure 6F*), suggesting a component of the learned response that scales with expectation. Notably, the same PC dendrites that respond to the cue after learning (n = 60/184) continue to respond to both the expected (p=$1.5 \times 10^{-4}$; one-tailed t-test; *Figure 6G*) and unexpected reward (p=$1.1 \times 10^{-15}$) after learning.

In contrast with area LS and Crus I, in naïve animals we observed a significant lick related response in Crus II as measured by the lick triggered average of Cspk activity during the ITI (p=$1.6 \times 10^{-5}$, n = 104; one-tailed t-test; *Figure 6H*). This result is consistent with previous reports suggesting that Cspk activity can be triggered by licking in this area (*Gaffield et al., 2016*; *Gaffield et al., 2018*; *Welsh et al., 1995*). Interestingly, however, these responses disappeared after learning (p=0.98; n = 136 dendrites, three mice). Moreover, after learning, the pre-reward timing of Cspks was independent of lick timing (Cspk latency on both early and late lick trials = 300 ms; *Figure 6I*), and Cspks maintained a similar rate following the cue (p=0.002; paired t-test; *Figure 6I*) regardless of lick timing. In addition, we also compared trials with the highest and lowest lick rates, and found the amount of licking also did not alter the timing of Cspk responses (high and low lick rate Cspk latency = 300 ms; *Figure 6J*), and produced Cspk responses with similar though statistically different amplitude prior to reward (p=$4.1 \times 10^{-5}$, paired t-test; *Figure 6J*). These results suggest that, while some activity can be driven by licking in naïve animals (*Figure 6H*), motor output alone is not the central determinant of Cspk activity in Crus II (*Figure 6H–L*).

Finally, we sought an additional approach to test whether movement contributes to the cue-associated, post-learning Cspk responses in LS, Crus I and Crus II. While we have demonstrated that Cspk responses do not correlate with licking, it remains possible that animals exhibit additional conditioned responses that are uncorrelated with licking. We thus used a piezoelectric sensor positioned under the animal to detect body movements during a subset of our post-learning imaging sessions (*Figure 7A*). This sensor is extremely sensitive to the smallest movements, even registering the animals' breathing (*Figure 7B*). Using this approach, we segregated trials according to the most and least movement on trial-by-trial basis, and found no relationship between the learned Cspk response and movement (High vs low movement Cspk responses, LS: p=0.13, n = 316 dendrites, five mice; Crus I: p=0.13, n = 27 dendrites, two mice; Crus II: p=0.10, n = 97 dendrites, three mice; *Figure 7C–E*). These data strongly support the hypothesis that the learned Cspk responses

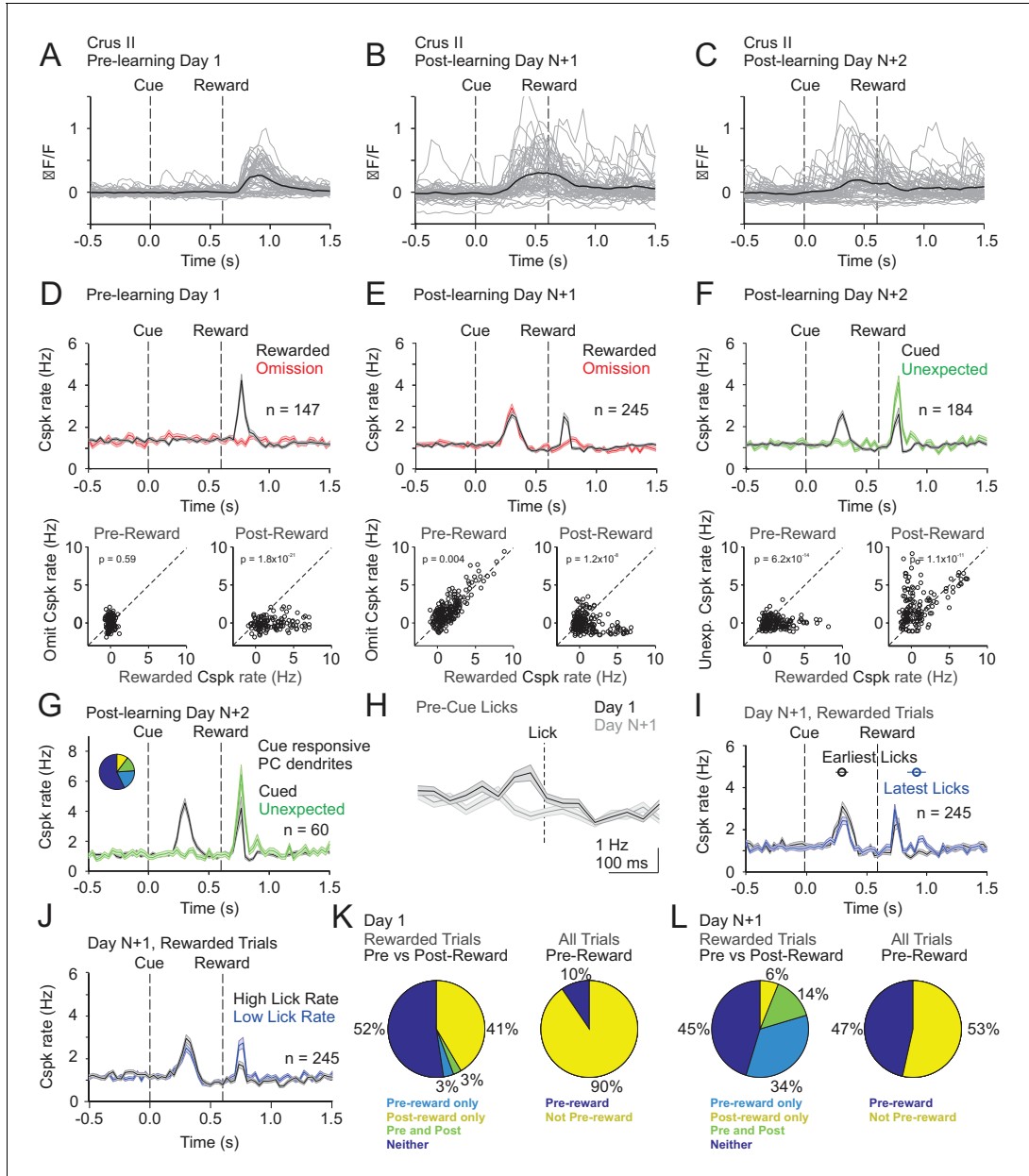

**Figure 6.** Cspk modulation in Crus II across learning. (A) ΔF/F timecourses from an example neuron measured via two-photon calcium imaging for the first day of training in naïve mice. Gray traces are the first 50 trials and the black trace is the average of those 50 trials. (B) Same as A), but on post-learning day N+1. (C) Same as (A,B) but for post-learning day N+2. (D) Top, mean cue-aligned PSTHs of Cspk rate on reward (black) and omission (red) trials for all PC dendrites on the first day of training in naïve mice. Bottom, summary scatterplot comparing the Cspk rate for individual PCs on rewarded vs omission trials in naïve mice. Spike rates were measured in a window preceding reward delivery (left) or immediately after reward delivery (right) (Materials and methods). (E) Same as (D) but for rewarded and reward omission trials on post-learning day N +1 (F) Same as (D,E) but for rewarded and unexpected reward trials on post-learning day N+2. (G) Mean cue-aligned PSTH for PC dendrites that exhibited Cspk responses to the visual cue on Day N+2. *Inset*: fraction of the total dendrites on Day N+2 that responded in the pre-reward window on rewarded trials only (light blue, 19%), the post-reward window on unexpected reward trials only (yellow, 10%), both pre and post reward windows on rewarded and unexpected reward trials respectively (green, 14%), and neither window (blue, 57%). (H) Mean lick-triggered PSTH for licks during the ITI in naïve animals. (I) Mean cue-aligned PSTHs for trained animals with rewarded trials segregated according trials with the earlies 1/4 of licks (black, 296 ± 52 ms from cue) and latest 1/4 of licks (blue, 914 ± 78 ms from cue). (J) Mean cue-aligned PSTHs for trained animals with rewarded trials segregated according to trials with the highest 1/4 of lick rates (black, 5.0 ± 0.2 Hz) and the lowest 1/4 of lick rates (blue, 2.5 ± 0.2 Hz) following reward delivery. (K) Fraction of all Crus II neurons which were responsive to specific task events on the first day of training for rewarded trials (left) and all trials (right). (L) Same as (K) but for training day N+1. Data points with horizontal error bars represent the mean lick timing ± SEM. For all PSTHs, shaded area represents ± SEM across dendrites.

DOI: https://doi.org/10.7554/eLife.46764.010

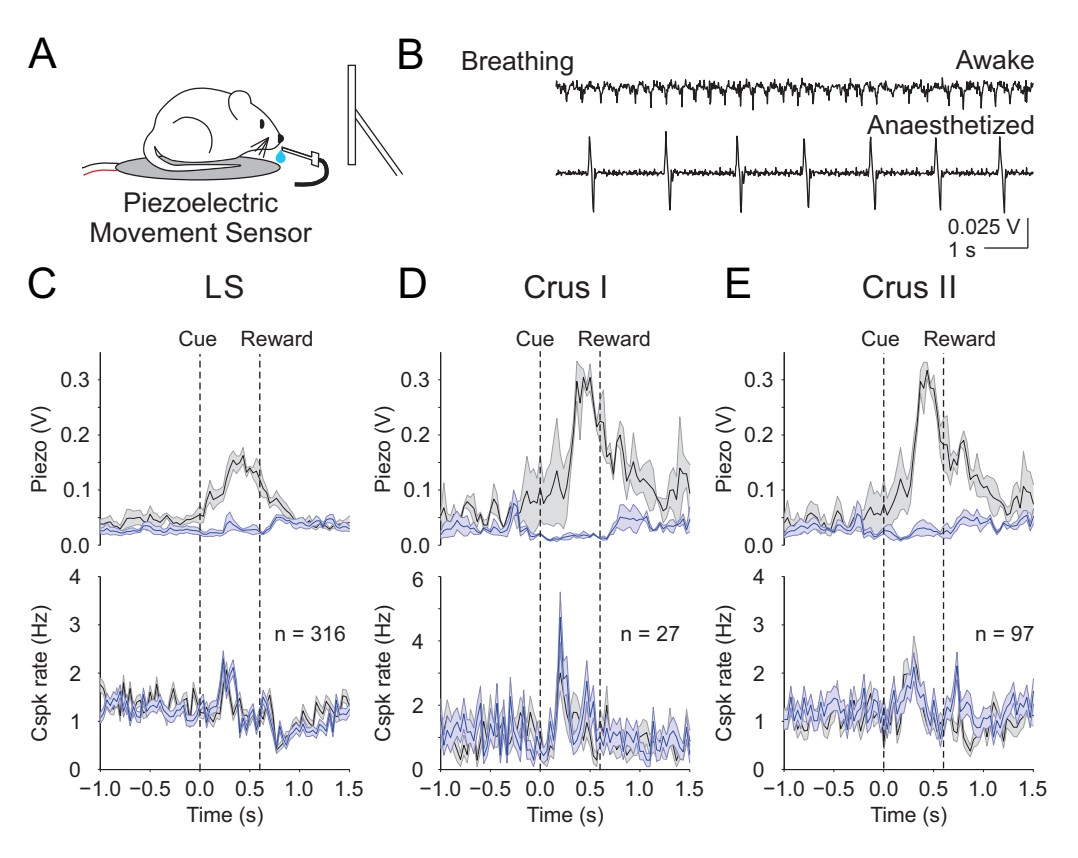

**Figure 7.** Body movements do not explain the learned reward predictive Cspks. (**A**) Piezoelectric movement sensor was used to measure movements during the behavior for a subset of animals. (**B**) movement traces corresponding to mouse breathing during awake (top) and anesthetized (bottom) conditions. Note that breathing becomes both slower and deeper (larger amplitude movements) under anesthesia. (**C**) Top, movement traces for the 10% of trials with the most (black) and least (blue) movement in the pre-reward window. Bottom, mean cue-aligned PSTHs from PCs in the lobule simplex corresponding to the subset of trials with the most (black) and least (blue) movement. (**D**) Same as B) but for Crus I. (**E**) Same as (**B,C**) but for Crus II. For all PSTHs, shaded area represents ± SEM across dendrites.
DOI: https://doi.org/10.7554/eLife.46764.011

demonstrated here are driven by the reward-predictive visual cue, and not reafference or sensory feedback related to movement.

## Discussion

We have used a classical conditioning paradigm of the kind typically studied to evaluate reward signaling in the VTA and striatum to reveal that climbing fibers provide a diverse set of learned reward prediction signals across the lateral cerebellum. Specifically, in LS and Crus II, Cspk responses emerged as a function of learning to a cue that predicted upcoming reward. In Crus I, while a sensory-evoked Cspk response to the visual cue was already evident in naïve animals, this response was enhanced once the cue had been associated with upcoming reward. Hence, in all three areas, there was a learned Cspk response related to reward prediction. Cspk responses also exhibited significant differences across areas. In LS, the response to expected reward was largely lost after learning, but was pronounced when reward was delivered unexpectedly. In Crus I, while there was no reward response in naïve animals, a Cspk response to unexpected reward emerged only after the animals established an expectation of reward delivery in response to the visual cue. Finally, in Crus II, reward responses persisted across learning, even after a separate Cspk response emerged to the reward-

predictive visual cue. The post-learning Cspk response to unexpected reward was enhanced, however, as compared to trials when reward was expected.

Together, these common, learned responses to a reward-predictive cue and to unexpected reward indicate that all three areas exhibit Cspk responses related to reward prediction. Such commonalities suggest a broad role for the lateral cerebellum in reward-based learning. The differences in responses, however, also suggest that learned changes in each area are overlaid on the unique sensory and motor Cspk signals pre-existing in each region. Thus, these data also indicate that the lateral cerebellum can harness other distinct sensory and motor signals across each area for different forms of sensorimotor learning, and perhaps to utilize alternate learning strategies that depend on task requirements.

Indeed, previous work supports the idea that LS, Crus I and Crus II may each have multiple functions, as they receive a diverse range of inputs, and have been implicated in both motor and non-motor behaviors. For example, climbing fibers in LS can carry somatosensory input related to forelimbs (*Apps and Garwicz, 2005*), but also exhibit responses consistent with reinforcement learning signals (*Heffley et al., 2018*; *Kostadinov et al., 2019*). Crus I climbing fibers can carry diverse sensory input related to visual, auditory and somatosensory stimuli (*Gaffield et al., 2019*; *Ju et al., 2019*), and Crus II climbing fibers can respond during various motor behaviors such as licking and whisker movement (*Brown and Raman, 2018*; *Gaffield et al., 2016*; *Gaffield and Christie, 2017*). Each of these areas also receives mossy fiber input that can convey both diverse sensory signals (*Apps and Hawkes, 2009*), as well as motor information (*Brown and Raman, 2018*; *Chen et al., 2016*; *Chen et al., 2017*). This combination of sensory and motor input has been suggested to enable integration that allows the lateral cerebellum to modulate voluntary behaviors though output to the neocortex (*Proville et al., 2014*). However, these lateral cerebellar regions have also been implicated in non-motor behaviors (*Badura et al., 2018*; *Carta et al., 2019*; *Deverett et al., 2018*; *Schmahmann et al., 2019*), and with cognitive deficits such as autism spectrum disorders (*Wang et al., 2014*). Such contributions to cognitive processes may be enabled by connectivity from these cerebellar regions to downstream targets such as the anterior cingulate cortex, prefrontal cortex, orbitofrontal cortex (*Badura et al., 2018*), hippocampus (*Hoshi et al., 2005*; *McAfee et al., 2019*) and striatum (*Hoshi et al., 2005*). In combination, this apparent complexity of connectivity and function suggests that correlations between neural activity and discrete behavioral parameters in absence of additional information about behavioral state or task goals may be insufficient to infer how these regions contribute to behavior.

In this study, cue and reward-associated Cspk responses could be clearly distinguished from motor responses due to licking or other movement in all three areas, as the timing or amount of licking did not correlate with Cspk responses, nor did overall movement as measured piezoelectric sensor. In a previous study, we identified a subset of lick-responsive PC dendrites after learning in area LS, and we concluded that such Cspk responses were learned and related to reward expectation rather than motor output per se (*Heffley et al., 2018*). Here, by imaging naïve animals, we were able to demonstrate no significant lick response in absence of learning in area LS and Crus I, consistent with previous reports indicating Cspk responses to licking are localized primarily to Crus II (*Gaffield et al., 2016*; *Gaffield et al., 2018*; *Welsh et al., 1995*). Also in agreement with these data, we did observe a significant Cspk response to licking in Crus II in naïve animals. Interestingly, however, once the association between visual cue and reward had been established, licking during the ITI when no reward was possible failed to produce Cspk activity. These data suggest that even in areas with motor related signals, Cspk responses are more complex than was previously appreciated. While earlier studies investigating Cspk responses to licking did not disambiguate the roles of motor output, reward, and reward expectation (*Bryant et al., 2010*; *Gaffield et al., 2016*; *Gaffield et al., 2018*; *Welsh et al., 1995*), we suggest that such efforts are crucial to understanding the role Cspk signals in cerebellar learning.

This study is part of a growing literature suggesting new roles for climbing fiber signals that are distinct from classical error-based supervised learning (*Heffley et al., 2018*; *Kitazawa et al., 1998*; *Kostadinov et al., 2019*; *Ohmae and Medina, 2015*; *Streng et al., 2017*). In a previous study, we identified Cspk activity consistent with an unsigned reinforcement learning rule in an operant conditioning task after learning (*Heffley et al., 2018*). In this study, we measured Cspk activity both before and after learning in a task with different requirements, and found responses that showed many similarities but also significant differences in Cspk activity. In both tasks, we observed Cspk

responses to an event that predicted upcoming reward. However, in the current study, Cspk activity was linked to a visual cue that predicted reward delivery, whereas previously we observed Cspk activity linked to an arm movement that was required to produce reward delivery. Indeed, a key difference between our previous operant task and the classical conditioning task studied here is that animals do not affect trial outcome with their motor output in this task. Specifically, while animals learn to lick earlier after learning in this task, a change in lick timing is not required for reward delivery, and there is no consequence for licks occurring with 'correct' or 'incorrect' timing (i.e. correctly or incorrectly matching the time of reward delivery). Hence, we speculate that observed differences in Cspk activity across tasks are related to the distinct sensory input and motor output that predicts reward delivery. In this task, the reward-predictive event was a visual cue alone. Previously, however, despite a visual cue that instructed arm movement, correct execution of the movement was required for reward delivery, and this action was therefore the necessary predictor of task outcome. In support of a role for task-specific movement requirements is shaping instructional signals, in the mesolimbic dopamine system it has been shown with a similar operant task that dopamine release is governed by correct movement initiation, and not simply reward prediction (*Syed et al., 2016*).

Another notable difference from our previous work is that we do not observe clear reward omission responses in this task. While it is unclear why such responses are absent in the current task, we again speculate that such responses may only be generated when movement is required to determine task outcome. In neither task, however, did we observe decreases in Cspk activity at the time when reward was expected on reward omission trials. Thus, we have not observed any evidence of signed, negative prediction errors in Cspk activity. These results contrast with a recent eyeblink conditioning study that used an aversive stimulus to drive learning (*Ohmae and Medina, 2015*), and further suggests a key role for task parameters in establishing the specific features of Cspk responses. Another possibility is suggested by the mesolimbic dopamine system, where neurons have different response profiles depending on their striatal projection target (*Menegas et al., 2017*; *Parker et al., 2016*). It thus remains possible that climbing fibers also exhibit an analogous target-specific response variability, and signal diverse reward-related information such as negative prediction errors in other regions of the cerebellum. Lastly, recent measurements of VTA dopamine neuron responses have revealed that increased firing to reward can persist after learning (*Coddington and Dudman, 2018*), similar to what we observe in Crus II. In those experiments, negative prediction errors did emerge eventually, but only very late in training, indicating that over-training can be required for omission responses to develop.

Finally, in this study we have measured Cspk responses across learning, thus revealing how Cspk activity shifts from reward-related responses to responses driven by reward predictive cues in the same animals. While we cannot rule out a contribution of sensory gating due to arousal, these data are most consistent with Cspks representing learned reward predictions. These data also suggest that the same climbing fibers can adjust their responses as a function of learning, and can signal diverse reward expectation information (i.e. both reward-predicting cues and unexpected reward in the same climbing fibers). These learned, reward-predictive climbing fiber signals are ideally suited to instruct both cerebellar learning and modulation of reward processing in downstream brain regions (*Chabrol et al., 2019*). In particular, a recent study has demonstrated that output from the lateral cerebellum via the deep cerebellar nuclei (DCN) forms a monosynaptic, excitatory connection onto dopamine neurons in the VTA (*Carta et al., 2019*). Activating these DCN projections is sufficient to positively modulate reward-driven behaviors, and these same projections are endogenously active under social conditions. These data strongly suggest that the cerebellum must contain, or have the ability to learn, information about rewarding stimuli. While speculative, such a model fits well with the observations made here. Specifically, we have shown Cspk instructional signals can become associated with stimuli that predict reward. If combined with contextual information from the mossy fiber pathway, such Cspk signals would be effective in instructing plasticity to depress Purkinje cell output and thereby enable DCN output to the VTA in response to any new, earlier reward-associated stimuli or actions. Hence, a key next step will be to evaluate the intersection of cerebellar learning with cerebellar output to the mesolimbic dopamine system during behavior. For the present, the patterns of Cspk activity demonstrated here provide key information about how instructional signals carried by cerebellar climbing fibers can change as a function of learning to signal reward expectation across the lateral cerebellum.

## Materials and methods

### Mice

All experimental procedures using animals were carried out with the approval of the Duke University Animal Care and Use Committee. All experiments were performed during light cycle using adult mice (p84-205) of both sexes, randomly selected from breeding litters. All mice were housed in a vivarium with normal light/dark cycles in cages with 1–5 mice. Imaging experiments were performed using Tg(PCP2-Cre)3555Jdhu mice (Jackson Labs, 010536; n = 34). A subset of these animals (n = 12) were the result of a genetic cross with Ai148(TIT2L-GC6f-ICL-tTA2)-D mice (Jackson Labs, 030328). We used two exclusion criteria for animals in this study: (1) poor recovery or other health concerns following surgical intervention or (2) missed virus injection, as determined by in vivo imaging and post-hoc histological analysis.

### Surgical procedures

3–10 hr prior to surgery, animals received dexamethasone (3 mg/kg) and ketoprofen (5 mg/kg). Surgical procedures were performed under anesthesia, using an initial dose of ketamine/xylazine (50 mg/kg and 5 mg/kg) 5 min prior to surgery and sustained during surgery with 1.0–2.0% isoflurane. Toe pinches and breathing were used to monitor anesthesia levels throughout surgeries. Body temperature was maintained using a heating pad (TC-1000 CWE Inc). Custom-made titanium headplates (HE Parmer) were secured to the skull using Metabond (Parkell). For imaging experiments, a 3 mm diameter craniotomy was made over the lobule simplex at approximately 1.4 mm lateral and 2.8 mm posterior to lambda or crus I and II at approximately 3.0 mm lateral and 4.3 mm posterior to lambda. Glass cover slips consisting of two 3 mm bonded to a 5 mm coverslip (Warner Instruments No. 1) with index matched adhesive (Norland No. 1) were secured in the craniotomy using Metabond. Buprenex (0.05 mg/kg) and cefazolin (50 mg/kg) were administered following surgery twice a day for two days. Following a minimum of 4 recovery days, animals were water deprived for 3 days, or until body weight stabilized at 85% of initial weight, and were habituated to head restraint (3–5 days) prior to behavioral training.

For imaging experiments involving viral expression of GCaMP (all non-Ai148 animals), the glass cover slip was removed following behavioral training, and mice were injected (WPI UMP3) with AAV1.CAG.Flex.GCaMP6f.WPRE.SV40 (UPenn vector core, titer = $1.48 \times 10^{13}$ or $7.60 \times 10^{12}$). 150 nL virus diluted 1:1-1:5 in ACSF was injected at a rate of 30 nl/min and a depth of 150 μm at 1–3 sites in dorsal lobule simplex. A new cranial window was then implanted, and imaging was performed beginning 10–14 days following injection. Transgenic mice crossed with the Ai148 mouse line received no injection and were given 10 days to recover from surgery before training.

### Behavior

During behavioral training, animals were head-fixed and placed in front of a computer monitor and reward delivery tube. Animals were trained to associate a visual cue with a saccharine reward. A high contrast vertical black and white grating was present at all times, including the inter-trial interval (ITI; 24–30 s), except when the reward cue was presented. The reward cue consisted of a horizontal grating, presented for 100 ms, and followed reward delivery after a 500 ms delay (600 ms from cue onset)(*Heffley et al., 2018*). Training sessions lasted for 62.1 ± 0.9 min and consisted of 130.1 ± 1.8 trials (n = 134 sessions). Imaging sessions lasted an average of 63.0 ± 1.2 min and consisted of 128.5 ± 2.5 trials (n = 71 sessions). For reward omission sessions, 20% of randomly determined trials were unrewarded. For unexpected reward sessions, no reward cue was presented on 20% of randomly determined trials. Behavioral parameters including cue presentation, reward delivery, and licking were monitored using Mworks (http://mworks-project.org) and custom software written in MATLAB (Mathworks) (*Heffley, 2019*). Learning was defined by stabilization of reaction times and miss rates, with required reaction times less than 650 ms and a miss rate below 15%. Across animals, 5.7 ± 0.4 training sessions (n = 27 mice imaged on both Day one and Day N+1) was required to meet these criteria. In a subset of experiments, (five mice in LS, three mice in Crus I and II), we used a piezoelectric sensor (C.B. Gitty, 41 mm 'jumbo' piezo) integrated into the behavioral apparatus to measure animal movement.

## Calcium imaging

### Mesoscale imaging

Single photon imaging was performed using a customized microscope (Sutter SOM) affixed with a 5x objective (Mitutoyo, 0.14NA) and CMOS camera (Qimaging, Rolera em-c$^2$). Excitation (470 nm) was provided by an LED (ThorLabs, M470L3), and data were collected through a green filter (520–536 nm band pass, Edmund Optics) at a frame rate of 10 Hz, with a field of view of 3.5 × 3.5 mm at 1002 × 1004 pixels.

### Two-Photon Imaging

Two-photon imaging was performed with a resonant scanning microscope (Neurolabware) using a 16x water immersion objective (Nikon CFI75 LWD 16xW 0.80NA). Imaging was performed using a polymer to stabilize the immersion solution (MakingCosmetics, 0.4% Carbomer 940). A Ti:Sapphire laser tuned to 920 nm (Spectra Physics, Mai Tai eHP DeepSee) was raster scanned via a resonant galvanometer (8 kHz, Cambridge Technology) at a frame rate of 30 Hz with a field of view of 1030 μm x 581 μm (796 × 264 pixels). Data were collected through a green filter (510 ± 42 nm band filter (Semrock) onto GaAsP photomultipliers (H10770B-40, Hamamatsu) using Scanbox software (Neurolabware). Experiments were aborted if significant z-motion was evident during recording. A total of 29 mice were used for imaging experiments across lobules in *Figures 2–6* (six mice for mesoscale imaging in LS; 16 mice for two-photon imaging in LS; seven mice for two-photon imaging in Crus I and Crus II (both lobules imaged in the same mice). In these experiments, the range of contributing dendrites to each experiment was as follows: LS: Day 1, 26–106 dendrites, 15 mice; Day N+1, 20–103 dendrites, 16 mice; Day N+2, 15–82 dendrites, six mice; Crus I: Day 1, 6–57 dendrites, six mice; Day N+1, 10–43 dendrites, seven mice; Day N+2, 10–57 dendrites, five mice; Crus II: Day 1, 18–40 dendrites, five mice; Day N+1, 24–78 dendrites, six mice; Day N+2, 10–46 dendrites, six mice. Five additional mice (separate cohort) were used to collect piezoelectric movement data on day N+1 for LS during imaging. For these experiments, the range of dendrites across experiments was 32–101. Piezoelectric movement data for Crus I and II came from a subset of the main cohort of animals described above.

## Data analysis and statistics

### Behavior

Analysis was performed using custom MATLAB code. Reaction times were defined as the first lick occurring 200–1000 ms after reward cue onset. Miss rate was defined as the percent of trials without a lick in the 1 s following cue presentation.

### Mesoscale imaging

Imaging analysis was performed using custom MATLAB code. Regions of interest (ROIs) were selected within LS according to the pattern of GCaMP expression. Window location and the lobule identity were identified according to folia patterns visible through the cranial window, landmarks recorded during surgery, and post-hoc histology. Baseline fluorescence (F) was measured on a trial-by-trial basis during the ITI as the mean F between 900 and 200 ms before trial initiation. Normalized fluorescence (ΔF/F) was calculated according to the cumulative activity within an ROI. ΔF/F latency was measured by first finding the peak ΔF/F response following cue onset averaged across trials. Next, the latency was measured as the time when mean ΔF/F trace achieved a maximum rate of change (the peak of the first derivative) in a 500 ms window preceding the identified peak ΔF/F response.

### Two-photon imaging

Motion in the X and Y planes was corrected by sub-pixel image registration. No data were excluded (after collection, see above) due to z-axis motion (assessed in each experiment by reviewing the trial averaged, cue aligned movie after X-Y motion correction). To isolate signals from individual PC dendrites, we utilized principal component analysis (PCA) followed by independent component analysis (ICA). Final dendrite segmentation was achieved by thresholding the smoothed spatial filters from ICA. A binary mask was created by combining highly correlated pixels (correlation coefficient >0.8)

and removing any overlapped regions between segmented dendrites. Notably, image segmentation using these criteria did not extract PC soma, which were clearly visible in some two-photon imaging experiments. Fluorescence changes (ΔF) were normalized to a window of baseline fluorescence (F) between 500 ms and 100 ms preceding trial initiation. One experiment was excluded from analysis based on single trial ΔF/F values that were more than double the amplitude of any other experiment, according to the rationale that any increases in Cspk activity during learning could be obscured by indicator saturation.

To extract events corresponding to Cspks, the first derivative of the raw fluorescence trace was thresholded at 1.7 standard deviations from baseline. Events above this threshold were variable in amplitude (*Figure 3—figure supplement 1*, also described previously by *Heffley et al., 2018*). Previous work has demonstrated that calcium transient amplitude variability reflects the properties of climbing fiber input, and not a combined climbing fiber and parallel fiber signal (*Gaffield et al., 2019*; *Gaffield et al., 2018*). All events separated by at least one frame were included. The pre-reward window was defined as the 600 ms between the cue and the reward delivery; the post-reward window was defined as the 600 ms following reward (or the time of expected reward). Peak spike rates for individual PC dendrites were defined by first identifying the peak spike rate in the mean peri-stimulus histogram (PSTH) across all PC dendrites within either the Pre- or Post-reward window. Next, this peak was used to define a 100 ms analysis window centered on the peak, to calculate spike rate. To identify PC dendrites responsive at the time of this peak, we selected only those PC dendrites with a peak response at least one standard deviation above their pre-cue baseline activity. To determine whether responses were significant across the population, we compared the responses of all PC dendrites at the time of the peak in the PSTH to a peak found in a 1200 ms window during the baseline and performed a one-tailed t-test across PC dendrites.

To identify PC dendrites that were responsive to task events (i.e. visual cue or reward), we selected PC dendrites with spike rates that were elevated one standard deviation above the pre-cue period for two consecutive frames in the pre- or post-reward windows. To identify PC dendrites that were suppressed following expected reward, we selected PC dendrites with spike rates that were suppressed one standard deviation below both their pre-cue and pre-reward firing rates for two consecutive frames during the post-reward window. These criteria selected a very small number of dendrites, which upon inspection, revealed no clear suppression associated with the time of expected reward (*Figure 3—figure supplement 3*).

Single licks during the ITI (separated by at least 250 ms from other licks) were used to generate lick triggered averages; only experiments with at least four licks were included for this analysis. Peaks in the lick triggered average were identified within a window spanning five frames centered on the lick. Significance was calculated by comparing the amplitude of this peak with two frames 250 ms before the lick and two frames 250 ms after the lick. To segregate trials according to lick times, on each trial we identified the first lick bout (at least 3 licks within 300 ms) following the cue, and separated trials into quartiles according to lick latency.

To track single PC dendrites across imaging sessions, we first independently generated pixel masks for PC dendrites from each field of view. We then motion registered the mean fluorescence image from the post-learning session to the pre-learning session, and applied the same pixel shifts from this registration to the segmented pixel masks from the post-learning condition. By overlaying the shifted post-learning pixel masks with the pre-learning masks, we assessed the overlap between imaging days. We employed a stringent criterion of greater than 50% overlap to increase the probability of accurately identifying the same PC dendrites across imaging sessions. With these criteria, we identified a group of PC dendrites likely corresponding to the same neurons before and after learning.

## Additional statistics

Data are presented as mean ± S.E.M., unless stated otherwise. Statistical tests were two-sided, except as specifically noted. Differences were considered statistically significant when $p < 0.05$. No correction for multiple comparisons was applied. No statistical methods were used to predetermine sample sizes. Data distribution was assumed to be normal but this was not formally tested. Data collection and analysis were not performed blind to the conditions of the experiments, but data

collection relied on automated measurements and subsequent analysis was based on code uniformly applied across experimental conditions.

## Acknowledgements

This work was supported by grants from the NIH NINDS (5R01NS096289-02) (CH) and (F31NS103425) (WH), the Sloan Foundation (CH), and the Whitehall Foundation (CH). We thank L Glickfeld for helpful discussions and input on calcium imaging approaches and analyses, and members of the Hull and Glickfeld labs for input and technical assistance throughout the project.

## Additional information

### Funding

| Funder | Grant reference number | Author |
| --- | --- | --- |
| National Institute of Neurological Disorders and Stroke | 5R01NS096289 | Court Hull |
| National Institute of Neurological Disorders and Stroke | F31NS103425 | William Heffley |
| Alfred P. Sloan Foundation | | Court Hull |
| Whitehall Foundation | | Court Hull |

The funders had no role in study design, data collection and interpretation, or the decision to submit the work for publication.

### Author contributions

William Heffley, Conceptualization, Data curation, Formal analysis, Funding acquisition, Validation, Investigation, Methodology, Writing—original draft, Writing—review and editing; Court Hull, Conceptualization, Formal analysis, Supervision, Funding acquisition, Methodology, Writing—original draft, Project administration, Writing—review and editing

### Author ORCIDs

William Heffley https://orcid.org/0000-0001-7733-7398
Court Hull https://orcid.org/0000-0002-0360-8367

### Ethics

Animal experimentation: All experimental procedures using animals were carried out with the approval of the Duke University Animal Care and Use Committee (protocol #A010-19-01).

### Decision letter and Author response

Decision letter https://doi.org/10.7554/eLife.46764.024
Author response https://doi.org/10.7554/eLife.46764.025

## Additional files

### Supplementary files

• Source data 1. Data organization files. Contains three files: A readme file that describes how the data are organized. "expNums_2P" and "expNums_LS_piezo" contain descriptive information about the two-photon experiments. expNums_2P is a 1x3 structure where each dimension is a different behavior condition (1=day1, 2=dayN+1, 3=dayN+2). The subfields contain information about which mice were imaged on each day, which lobules were imaged, and how many neurons were in the field of view. The file expNums_LS_piezo contains the same information but only for the lobule simplex datasets collected with the piezo movement data.
DOI: https://doi.org/10.7554/eLife.46764.012

• Source data 2. Widefield data.
DOI: https://doi.org/10.7554/eLife.46764.013

• Source data 3. Two photon LS piezo data.
DOI: https://doi.org/10.7554/eLife.46764.014

• Source data 4. Two photon C1.
DOI: https://doi.org/10.7554/eLife.46764.015

• Source data 5. Two photon C2.
DOI: https://doi.org/10.7554/eLife.46764.016

• Source data 6. Two photon LS data. Day_N1 (1).
DOI: https://doi.org/10.7554/eLife.46764.017

• Source data 7. Two photon LS data. Day_N1 (2).
DOI: https://doi.org/10.7554/eLife.46764.018

• Source data 8. Two photon LS data. Day_1 (1).
DOI: https://doi.org/10.7554/eLife.46764.019

• Source data 9. Two photon LS data. Day_1 (2).
DOI: https://doi.org/10.7554/eLife.46764.020

• Source data 10. Two photon LS data. Day_N2.
DOI: https://doi.org/10.7554/eLife.46764.021

• Transparent reporting form DOI: https://doi.org/10.7554/eLife.46764.022

## Data availability

Datasets supporting the findings of this study are ~50 GB per experiment, and are therefore available through a request to the corresponding author. Processed data have been provided for each figure, and analysis code has been placed in GitHub (https://github.com/Glickfeld-And-Hull-Laboratories/Heffley_Hull_2019_eLife; copy archived at https://github.com/elifesciences-publications/Heffley_Hull_2019_eLife).

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
