## [Decision Letter]

Thank you for submitting your article "Classical conditioning drives learned reward prediction signals in climbing fibers across the lateral cerebellum" for consideration by *eLife*. Your article has been reviewed by three peer reviewers, and the evaluation has been overseen by a Reviewing Editor and Ronald Calabrese as the Senior Editor. The following individuals involved in review of your submission have agreed to reveal their identity: Talia Lerner (Reviewer #1); Isabel Llano (Reviewer #2); Kamran Khodakhah (Reviewer #3).

The reviewers have discussed the reviews with one another and the Reviewing Editor has drafted this decision to help you prepare a revised submission.

Summary:

Through the study of climbing fibre calcium activity before and after a learning task, this work aims to study how cerebellar learning contributes to the acquisition of a reward-driven behaviour. The work is timely, since recent evidence indicates that the cerebellar circuit intervenes in behaviours that have an "emotional" and/ or "cognitive" content. The authors used single and two photon calcium imaging, as a proxy for neuronal activity, to examine the response of the cerebellar cortex and individual Purkinje cells in behaviors typically used to explore classical conditioning and reward processing in the reward circuitry. In their 2P imaging studies, they explored complex spike-evoked calcium signals in the dendrites of Purkinje cells. Depending of the specific cerebellar region imaged, calcium signals from complex spikes (Cspk) are active either while the animal learns to associate a visual cue with a reward or once the association is accomplished. The key conclusion is that the Cspks primarily signaled reward prediction, rather than reward prediction error, and hence the Cspk is not functioning according to the predictions of a classical "temporal difference learning model".

Overall, this is rigorous and interesting work. The presentation is generally clear and the conclusions supported by the data and pertaining analysis, with recognition of recent publications pertaining to the aim of the present work. The reviewers are enthusiastic, but would like to see a few issues addressed.

Essential revisions:

1) The reviewers acknowledged the technical difficulty of these experiments, yet felt the number of animals seemed insufficient in some cases, with less than three animals in some sub groups. Increasing the number of animals per group to have meaningful statistical power when all the data from each animal are considered to be an n=1 would greatly strengthen the manuscript.

Moreover, although information on number of mice used for different regions is given in Materials and methods, there are sections of Results in which it is unclear how many animals are being used in each data set. For example, in the subsection “An appetitive classical conditioning paradigm to investigate climbing fiber responses in the lateral cerebellum”, n = 24, from how many mice?. In the first paragraph of the subsection “Cspk responses in LS signal reward predictions that emerge with learning”, n = 1040 dendrites, from how many mice? In Figures 3, 5, 6 and Figure 3—figure supplements 1 and 2, the number of repetitions for the averaged traces is given in the panels as n = XX, but from how many dendrites and from how many mice is not stated in the text or Figure legends. For Figure 4 there is no mention of number of repetitions or of the number of animals. For each statistical comparison, the animal number and, when it applies, number of dendrites per animal should be specified in the corresponding section of the main text and/or in the figure legend. This is most important when discussing data from subsets of PC dendrites as in the subsection “The same climbing fibers can respond to reward and a reward-predictive visual cue”, since such results may be derived from a smaller number of animals.

2) The possibility that the Cspk responses associated with reward prediction are actually driven by licking or other movements made by the mice in anticipation of reward should be more rigorously addressed.

More extensive analysis of licking would be helpful. The strongest evidence that the Cspk responses are not licking-related was that the shifts in timing for the Cspks and the licks with learning were not aligned. The authors also indicate that the Cspk latencies are not substantially different (30 ms) for trials with the fastest vs. slowest lick latencies. The lick latencies for those samples should also be given, so that the reader can evaluate whether there was sufficient variability in the licking for the lack of variability in the Cspk latency to provide evidence for a dissociation between the two. In addition, although the lick-triggered averages are informative, perhaps isolated licks in the ITI aren't representative of effects that could be observed during more intense licking bouts, hence it may be helpful to look at Cspk latencies in trials with more vs. less intense licking, in addition to the latency analysis.

A key issue is that it is highly likely that the mice are making a variety of other movements besides licking when they are anticipating a reward, which could potentially drive the Cspk responses. Jorntell et al., 2000 showed short and long-latency Cspk responses to somatosensory stimulation in similar regions of the cerebellum in rats. This might explain why the Cspk responses look more like reward prediction than reward prediction error-- one would not necessarily expect opposite movements (or opposite movement-related climbing fiber responses) for reward cues and omitted rewards. This potential confound should be addressed, ideally with additional experiments or analyses, but at the very minimum with explicit acknowledgement and discussion.

Finally, the authors should consider whether their observations could be conceived as a general arousal effect rather than specific to reward processing.

The reviewers recognize that it is difficult to control for all of the possible movements an animal might make when anticipating a reward. Yet there was also concern about the confusion that could be created by labeling signals as reward without more rigorously ruling out potential confounds such as movements or motor reafference.

3) There are several places where fuller representation and/or analysis of the data would be helpful.

a) PC dendrites responding to the visual cue after training also respond to unexpected reward in many cases, though not all. Are these the same PC dendrites that responded to reward initially in each area? Are there PC dendrite that respond to unexpected reward but not cues? What are the differences in distributions of how many PC dendrites represent each thing across the different regions? E.g. Do unexpected reward responses only emerge with training in Crus I or is that also happening in subsets of cells in the other areas? It could be helpful to see a statistical criteria established for which dendrites significantly respond to each stimulus, then put those in color on the scatter plots, and maybe have pie charts to show distributions of responsive neurons in each of the different cerebellar areas.

b) The amplitude and the variability of the calcium signals is not discussed. Statistics on the mean peak DF/Fo value for pooled data before and after learning would be useful. Even is this parameter is not used in the present work, it may be of value for other studies. In the figures, the sem of the averaged calcium traces could be shown along with the mean trace. Moreover, the authors should address the issue of how the non-linearity of the calcium signals may affect the interpretation of Results.

c) The authors also find that in a number of Purkinje cells the complex spikes change the nature of the information that encode. The reviewer felt that these data were not analyzed as extensively as possible (perhaps because of the relatively small number of animals and identified cells), to explore whether there is anything special or different about these Purkinje cells in terms of spatial distribution (lobules, sub lobules, or zebrin bands), and temporal progression.

4) The Discussion should be expanded to consider how previous knowledge of the function of these regions of the cerebellum can inform the interpretation of the current results.

What is the significance of LS, Crus I and Crus II? What were the previously known distinctions between them that caused the authors to compare and contrast?

In Carta et al., 2019, it's not clear where within the cerebellum monosynaptic inputs to the VTA arise. Would the signals the authors observe here actually be setup to control output to the VTA? This seems like a key question to resolve a bit more given the stated purpose and takehome message of the work.

Wagner et al., 2017 looked at granule cells over simplex, V, VIa, VIb. How does this align with the authors' observations here? Would the granule cells Wagner recorded also send inputs to the same PC dendrites the authors are interested in?

The main difference between this manuscript and the authors' previously published work (that is emphasized by the authors themselves) is the difference between Pavlovian vs. instrumental learning. This is an important distinction, but raises the question of whether the implication of this work (coupled with the authors' evidence that much of this reward signaling is motor-independent) is that the Cspk responses observed are used primarily for driving motor-independent Pavlovian associations? Are they interfacing with motor information in some other part of the circuit?

---

## [Author Response]

Essential revisions:1) The reviewers acknowledged the technical difficulty of these experiments, yet felt the number of animals seemed insufficient in some cases, with less than three animals in some sub groups. Increasing the number of animals per group to have meaningful statistical power when all the data from each animal are considered to be an n=1 would greatly strengthen the manuscript.Moreover, although information on number of mice used for different regions is given in Materials and methods, there are sections of Results in which it is unclear how many animals are being used in each data set. For example, in the subsection “An appetitive classical conditioning paradigm to investigate climbing fiber responses in the lateral cerebellum”, n = 24, from how many mice?. In the first paragraph of the subsection “Cspk responses in LS signal reward predictions that emerge with learning”, n = 1040 dendrites, from how many mice? In Figures 3, 5, 6 and Figure 3—figure supplements 1 and 2, the number of repetitions for the averaged traces is given in the panels as n = XX, but from how many dendrites and from how many mice is not stated in the text or Figure legends. For Figure 4 there is no mention of number of repetitions or of the number of animals. For each statistical comparison, the animal number and, when it applies, number of dendrites per animal should be specified in the corresponding section of the main text and/or in the figure legend. This is most important when discussing data from subsets of PC dendrites as in the subsection “The same climbing fibers can respond to reward and a reward-predictive visual cue”, since such results may be derived from a smaller number of animals.

We agree that animal numbers were previously low for our Crus I and II experiments, and that our reporting of animal and dendrite numbers was not sufficiently detailed. Thus, we have now performed additional experiments in Crus I and II to bolster the number of animals and dendrites from these sessions (each of these experiments now has a minimum of 5 animals). Moreover, within the Results section, each experiment now references both the number of animals and the number of dendrites contributing to each measurement. In addition, we have also added the range of dendrites per animal contributing to each experiment in the Materials and methods section (subsection “Two-Photon Imaging”).

2) The possibility that the Cspk responses associated with reward prediction are actually driven by licking or other movements made by the mice in anticipation of reward should be more rigorously addressed.More extensive analysis of licking would be helpful. The strongest evidence that the Cspk responses are not licking-related was that the shifts in timing for the Cspks and the licks with learning were not aligned. The authors also indicate that the Cspk latencies are not substantially different (30 ms) for trials with the fastest vs. slowest lick latencies. The lick latencies for those samples should also be given, so that the reader can evaluate whether there was sufficient variability in the licking for the lack of variability in the Cspk latency to provide evidence for a dissociation between the two. In addition, although the lick-triggered averages are informative, perhaps isolated licks in the ITI aren't representative of effects that could be observed during more intense licking bouts, hence it may be helpful to look at Cspk latencies in trials with more vs. less intense licking, in addition to the latency analysis.

We have performed a new analysis to evaluate a possible role for licking intensity in our Cspk responses: Specifically, as suggested, we have selected trials with the highest and lowest lick rates (measured starting at the time of the visual cue), and compared Cspk responses across these trial types. We find no relationship between the intensity of licking and the pre-reward Cspk responses in any of the three imaged lobules (Figures 3, 5 and 6, panel J). For these experiments, we have reported lick rates in the figure legends to illustrate the separability of our “high” and “low” lick rate categories. We have also added quantification of lick timing for the “early” and “late” lick categories to the figure legends as requested. We note that lick timing was previously plotted on panel F in Figures 3, 5 and 6 with S.E.M. However, these points were somewhat small, and may have been difficult to discern. We have thus made the lick latency points larger, and labeled them on the figure to better illustrate the separability of “early” vs. “late” lick categories (Figures 3, 5, and 6 panel I).

A key issue is that it is highly likely that the mice are making a variety of other movements besides licking when they are anticipating a reward, which could potentially drive the Cspk responses. Jorntell et al., 2000 showed short and long-latency Cspk responses to somatosensory stimulation in similar regions of the cerebellum in rats. This might explain why the Cspk responses look more like reward prediction than reward prediction error-- one would not necessarily expect opposite movements (or opposite movement-related climbing fiber responses) for reward cues and omitted rewards. This potential confound should be addressed, ideally with additional experiments or analyses, but at the very minimum with explicit acknowledgement and discussion.

This is an extremely important point, and one that we felt required a new experimental approach to more rigorously test whether some other conditioned movement (apart from licking) might explain the pre-reward Cspks that occur after learning. Thus, we have integrated a piezoelectric movement sensor into the animals’ behavioral apparatus. This sensor is exquisitely sensitive to even the smallest movements (for example, we find that it provides a reliable measure of mouse breathing). This setup has allowed us to collect a continuous record of animal movement during our experiments. Using this system, we have performed new imaging experiments and added the results to a new Figure 7 (Results subsection “Crus I and Crus II exhibit reward-related Cspk responses that are distinct from area LS”, last paragraph). When trials are sorted according to the most and least movement, we find that the learned, pre-reward Cspk response is independent of movement.

Finally, the authors should consider whether their observations could be conceived as a general arousal effect rather than specific to reward processing.

It would indeed be interesting if Cspk responses could be gated by the arousal state of the animal. However, in our view this is an unlikely explanation for the changes we have measured in prereward Cspks across learning, as animals are not disengaged on the first day of training. On Day 1, naïve water deprived animals are significantly aroused when they are receiving bouts of unexpected reward for the first time. However, because we do not have any specific measure of arousal such as pupil tracking to support this claim, we have added a statement to the Discussion indicating the possibility of arousal in contributing to the Cspk responses measured here (Discussion, last paragraph).

The reviewers recognize that it is difficult to control for all of the possible movements an animal might make when anticipating a reward. Yet there was also concern about the confusion that could be created by labeling signals as reward without more rigorously ruling out potential confounds such as movements or motor reafference.

By incorporating a piezoelectric movement sensor into our measurements, we have endeavored to provide as rigorous a test as we could devise to evaluate this issue. In our view, the results of these experiments, along with more extensive quantification of licking, make reafference or sensory feedback from movement an unlikely explanation for the learned Cspk responses we have measured.

However, we agree that conclusively disambiguating sensory and movement signals is extremely challenging, and it is always possible that there are movements we still cannot detect. We hope that our new experiments and discussion of movement (Discussion) will provide a sufficiently balanced perspective, and serve to mitigate potential confusion in interpreting our data.

3) There are several places where fuller representation and/or analysis of the data would be helpful.a) PC dendrites responding to the visual cue after training also respond to unexpected reward in many cases, though not all. Are these the same PC dendrites that responded to reward initially in each area? Are there PC dendrite that respond to unexpected reward but not cues? What are the differences in distributions of how many PC dendrites represent each thing across the different regions? E.g. Do unexpected reward responses only emerge with training in Crus I or is that also happening in subsets of cells in the other areas? It could be helpful to see a statistical criteria established for which dendrites significantly respond to each stimulus, then put those in color on the scatter plots, and maybe have pie charts to show distributions of responsive neurons in each of the different cerebellar areas.

We agree that more extensive characterization of Cspk responses on different trial types before and after learning would be useful. Thus, we have added several pie charts to Figures 3, 5 and 6 to illustrate how PC dendrites respond to: The visual cue and unexpected reward after learning (inset for panel G), and the visual cue and reward delivery before and after learning (panels K and L). We have also added a new panel to Figure 4J to directly show firing rates of the same neurons to reward on Day 1 and the visual cue after learning. Statistical criteria for responsivity are described in the Materials and methods.

b) The amplitude and the variability of the calcium signals is not discussed. Statistics on the mean peak DF/Fo value for pooled data before and after learning would be useful. Even is this parameter is not used in the present work, it may be of value for other studies. In the figures, the sem of the averaged calcium traces could be shown along with the mean trace. Moreover, the authors should address the issue of how the non-linearity of the calcium signals may affect the interpretation of Results.

We have now added raw DF/F traces from example cells (taken from the same animal within each figure) to Figures 3, 5 and 6 showing the first 50 rewarded trials and the average of those trials. To visualize the variability in DF/F, we have added a supplementary figure (Figure 3—figure supplement 1) quantifying the dF/F amplitude in the time windows before and after reward for both Pre and Post-learning sessions across all areas for all measured dendrites.

With the analysis approaches we have used, it is not clear what effect calcium transient or indicator non-linearities would have on our interpretations. By thresholding the DF/F traces to generated binarized spiking, we have restricted our interpretation to whether or not an event occurred at any given timepoint, regardless of variation in amplitude. Cspk Ca transients are extremely large, and well separated in time on a single trial basis (as documented previously in Heffley et al., 2018). Thus, there is little chance of missing events due to differences in indicator expression across cells or working in the sublinear range of the indicator. Moreover, previous work has suggested that there is little reason to suspect indicator saturation is a concern for brief bursts of complex spikes in awake mice (Gaffield et al., 2019). Certainly, responses diminish during bursts, and we cannot detect unique events unless they are separated by at least one frame (30 ms). However, these issues do not significantly complicate our approach of binary event detection, nor should they significantly alter our measures of mean event timing in these experiments.

Regarding calcium transient non-linearities, recent work has made clear that no contribution of parallel fiber input can be detected with these approaches (parallel fiber input does not enhance the Cspk response unless inhibition is blocked, and parallel fiber input cannot be detected when there is no Cspk) (Gaffield et al., 2018, Gaffield et al., 2019). We are well aware that combined Cspk and parallel fiber input can produce calcium non-linearities under some conditions, but it would appear that such non-linearities that have been measured in vitro at the level of spines may either not occur broadly across the dendritic shaft, or not be detectable with the global, in vivo dendritic calcium measurements that we and others have performed (at least when synaptic inhibition from MLIs remains intact). We have added a statement of rationale regarding our thresholding approach and its relationship to possible calcium transient non-linearities to the Materials and methods.

c) The authors also find that in a number of Purkinje cells the complex spikes change the nature of the information that encode. The reviewer felt that these data were not analyzed as extensively as possible (perhaps because of the relatively small number of animals and identified cells), to explore whether there is anything special or different about these Purkinje cells in terms of spatial distribution (lobules, sub lobules, or zebrin bands), and temporal progression.

A more detailed, within lobule examination of variability across the medio-lateral cerebellar axis is not practical with the current dataset. We do not have a way to reveal zebrin bands during imaging, or label them post-hoc in a way that enables registration to our imaging ROIs. Even in absence of ground truth established by a histological feature such as zebrin staining, the numbers of isolated neurons per experiment in our dataset are not sufficient for further breakdown of subareas (nor is the field of view large enough for such comparisons spanning greater than 200 μm microzones). Such within lobule measurements would indeed make a compelling future study, but lies beyond the scope of what we can accomplish here.

4) The Discussion should be expanded to consider how previous knowledge of the function of these regions of the cerebellum can inform the interpretation of the current results.What is the significance of LS, Crus I and Crus II? What were the previously known distinctions between them that caused the authors to compare and contrast?

The rodent lateral cerebellum remains poorly understood in terms of both function and detailed anatomical projections. Each of these areas has been implicated in non-motor function and cognitive processes, and therefore our motivation was simply to assess how climbing fiber driven instructional signals in these areas respond in a reward-association task that does not require motor output to influence task outcome. Ideally, we would have also tested even more lateral areas, but our imaging approach only provides access to the lateral areas we have tested.

We have now included a new section in the Discussion to highlight what is known about these areas and better contextualize our findings.

In Carta et al., 2019, it's not clear where within the cerebellum monosynaptic inputs to the VTA arise. Would the signals the authors observe here actually be setup to control output to the VTA? This seems like a key question to resolve a bit more given the stated purpose and takehome message of the work.

It is our working hypothesis that a climbing fiber signal driven by a reward-predictive cue could serve to drive Purkinje cell LTD for any temporally-associated context carried by the parallel fibers. If so, this would enable cerebellar output (DCN spiking) in response to a context where reward is likely, and accordingly generate an excitatory input to the VTA like those measured by Carta et al. We have described this model in the last paragraph of the Discussion. However, in absence of more direct evidence for such a model, we cannot push further than this general hypothesis with support from our current data.

Wagner et al., 2017 looked at granule cells over simplex, V, VIa, VIb. How does this align with the authors' observations here? Would the granule cells Wagner recorded also send inputs to the same PC dendrites the authors are interested in?

We view the signals recorded by Wagner as additional information that can be pooled by the granule cells to form complex sensory, motor, and contextual representations that can be learned by Purkinje cells. We view climbing fiber signals differently; namely, because they are instructional, we view them as a way to infer the learning rules implemented by the cerebellum. Our work suggests an extension to the error-based supervised learning rules that have typically been described in other types of behaviors. In other words, we view climbing fiber signals as way to understand how the cerebellum learns, and the granule cell responses as a way to understand what can be learned.

Importantly, in Wagner et al., reward prediction was one of many features the granule cells encoded, along with other task features (such as motor information). We thus find it unlikely that Purkinje cells would receive a confluence of the same information from granule cells and climbing fibers in such tasks. The Wagner data suggests that reward information is pooled with other sensorimotor information in the granule cell layer in order to establish unique contexts that can be learned by Purkinje cells (perhaps according to the reinforcement learning rules implied by our experiments). Apart from such distinctions, we do not have meaningful hypotheses about what it would mean if the Wagner granule cells impinge on the Heffley Purkinje cells, or what features of cerebellar learning would be conferred if the Wagner granule cells impinge on Heffley Purkinje cells in LS vs. Crus I vs. Crus II.

The main difference between this manuscript and the authors' previously published work (that is emphasized by the authors themselves) is the difference between Pavlovian vs. instrumental learning. This is an important distinction, but raises the question of whether the implication of this work (coupled with the authors' evidence that much of this reward signaling is motor-independent) is that the Cspk responses observed are used primarily for driving motor-independent Pavlovian associations? Are they interfacing with motor information in some other part of the circuit?

These are great questions, and we do not have experimental evidence to answer them at this point. The differences in Cspk responses between this task and our previous operant task occur primarily when reward is omitted. It is possible that the difference between tasks on omission trials reflects a difference between signaling reward prediction and reward prediction error, and that reward prediction error is only necessary in the motor domain. However, it also remains possible that we simply did not observe reward prediction error signals on omission trials in this study because we measured responses immediately after learning. We are now beginning to test whether robust omission responses depend on training history or duration of training, as has been suggested by the dopamine literature. To fully explore these issues will require a future study, as this is a ‘next-step’ question that will require significant new experimental effort.

There are also many features of Cspk signaling that are shared between paradigms, and we speculate that the learned, reward-predictive Cspk responses that are common between this task and our previous task could serve to instruct LTD and thus cerebellar output for any granule cell input that is co-active. Such inputs could carry motor or non-motor contextual information. Thus, how these signals are used would depend on what is carried by the granule cells, as well as what output pathway is engaged (where the co-active granule cell inputs occur). We thus think it is possible that the learned, cue-predictive Cspk responses could enable both motor and non-motor associations that are reward predictive. In our view, the wide distribution of these signals across the lateral cerebellum supports such a hypothesis. We have now amended the Discussion to highlight what is known about such inputs and outputs across the lateral cerebellum, and to provide a basis for what these climbing fibers signals might enable.